# Reinforcement Learning-Based Resource Allocation Scheme of NR-V2X Sidelink for Joint Communication and Sensing

**DOI:** 10.3390/s25020302

**Published:** 2025-01-07

**Authors:** Zihan Li, Ping Wang, Yamin Shen, Song Li

**Affiliations:** College of Information Science and Technology, Donghua University, Shanghai 201620, China; 2221956@mail.dhu.edu.cn (Z.L.); 1219130@mail.dhu.edu.cn (Y.S.); 1239125@mail.dhu.edu.cn (S.L.)

**Keywords:** joint communication and sensing (JCS), NR-V2X sidelink, rescource allocation, radar sensing, Q-learning

## Abstract

Joint communication and sensing (JCS) is becoming an important trend in 6G, owing to its efficient utilization of spectrums and hardware resources. Utilizing echoes of the same signal can achieve the object location sensing function, in addition to the V2X communication function. There is application potential for JCS systems in the fields of ADAS and unmanned autos. Currently, the NR-V2X sidelink has been standardized by 3GPP to support low-latency high-reliability direct communication. In order to combine the benefits of both direct communication and JCS, it is promising to extend existing NR-V2X sidelink communication toward sidelink JCS. However, conflicting performance requirements arise between radar sensing accuracy and communication reliability with the limited sidelink spectrum. In order to overcome the challenges in the distributed resource allocation of sidelink JCS with a full-duplex, this paper has proposed a novel consecutive-collision mitigation semi-persistent scheduling (CCM-SPS) scheme, including the collision detection and Q-learning training stages to suppress collision probabilities. Theoretical performance analyses on Cramér–Rao Lower Bounds (CRLBs) have been made for the sensing of sidelink JCS. Key performance metrics such as CRLB, PRR and UD have been evaluated. Simulation results show the superior performance of CCM-SPS compared to similar solutions, with promising application prospects.

## 1. Introduction

In the future, vehicles on the road will need to frequently exchange information with surrounding vehicles, pedestrians, and road traffic infrastructure, which drives the development of Vehicle-to-Everything (V2X) technology. Today’s vehicles have transformed from traditional vehicles into intelligent vehicles. Through the V2X network, there is potential to achieve an Advanced Driver Assistance System (ADAS), improving driving safety and comfort. Recently, the latest beyond 5G and 6G standards have introduced new requirements for V2X, including enhanced demands for sensing accuracy, precision, and resolution, alongside the existing communication criteria of latency, reliability, capacity, and coverage [1]. Therefore, the joint communication and sensing (JCS) system that utilizes a signal to simultaneously achieve two functions has attracted much attention.

In previous systems, communication and radar sensing were separate systems using different frequencies and hardware resources. However, with increasingly scarce spectrum resources, there is a need for more efficient utilization of spectrum resources by communication and radar systems. As the bandwidth of commercial communication systems increases, coexistence with various existing radar systems is anticipated, leading to the development of the JCS concept [2]. JCS can provide integrated and collaborative gains for future systems [3]. On the one hand, sharing spectrum and hardware resources can lead to high resource utilization efficiency. On the other hand, the sensing function can assist communication in obtaining more accurate channel estimation models, which are beneficial for beamforming and spectrum resource management.

3GPP Release 16 has established standards for vehicle sidelink communication based on the 5G-NR PC5 air interface, enabling vehicles to communicate directly without the assistance of gNB [4], as illustrated in Figure 1. Sidelink is beneficial for reducing latency and improving communication. In addition, sidelink signals can also be used for near-field positioning, range sensing, and distance measurements [5], thereby complementing or enhancing positioning systems that may be limited by obstacles or other factors, such as network-based positioning or the Global Navigation Satellite System (GNSS). Therefore, the V2X sidelink JCS system has significant development potential.

However, due to limited available bandwidth, and without the assistance of a base station, there is a conflicting requirement between radar and communication in spectrum resource utilization. Radar accuracy requires large bandwidth occupancy, which reduces available resources in the resource pool, increases the resource collision probability, and thereby affects the performance of communication. The issue of resource collision in the sidelink scenario is related to resource allocation schemes. Therefore, a flexible and robust resource allocation scheme is crucial for mitigating resource pool conflicts.

Traditional sidelink resource allocation schemes are divided into dynamic allocation and sensing-based semi-persistent scheduling (SB-SPS). SB-SPS is widely used for sidelink resource allocation due to its better reliability and latency [6]. The SB-SPS scheme firstly senses the channel quality and selects the available candidate resources and does not select a new resource for the next transmission until the retransmission counter (RC) returns to zero in order to avoid packet collision during the sidelink communication access. There are also numerous studies concerning modified SPS schemes to improve low-latency communication [7,8]. However, the existing research has rarely studied the impacts of packet collision on sidelink sensing performance. According to the theoretical model of Cramér–Rao Lower Bound (CRLB), the resultant SINR of the echo due to collisions will significantly impact the sensing performance of the JCS sidelink. Especially, in high-density scenarios, the probability of selecting the same resource for different vehicles quickly rises, resulting in the deterioration of both communication and sensing performances. In fact, obtaining knowledge about the dynamic echo channel state of JCS is still a big challenge and difficult to resolve. Therefore, it is rational for this paper to study the consecutive collision problem related to echo together with transmission signals of the JCS sidelink. With the advancements in self-interference (SI) technology in recent years [9], simultaneous transmission and reception on the same frequency band with in-band full-duplex (FD) transceivers have become feasible, offering hope for the implementation of JCS systems in the sidelink. Additionally, the powerful sensing capabilities of full-duplex bring collision detection functionality, creating new opportunities for enhancing sidelink resource allocation schemes.

Moreover, in response to the high dynamics of V2X traffic density and network load, inspired by references [10,11], reinforcement learning and other intelligent algorithms can be employed to optimize resource allocation, ensuring the stability and accuracy of communication and sensing tasks.

Inspired by the above, this paper focuses on high-positioning accuracy, low-latency and high-reliability in the 5G NR-V2X sidelink JCS system. By studying the comprehensive impact of interference due to consecutive collisions, we propose a reinforcement learning-based collision mitigation resource allocation scheme (CCM-SPS). Specifically, this scheme employs JCS full-duplex collision detection and reinforcement learning to optimize traditional SB-SPS parameters, mitigating performance degradation from consecutive collisions. Furthermore, the impact of varying vehicle density and packet sizes on JCS performance in dynamic vehicular networks is discussed. Finally, the effectiveness of the proposed scheme in enhancing overall performance is validated using a V2X sidelink system-level simulator.

The main contributions of this work can be summarized as follows:In order to address the conflicting requirement between sensing accuracy and communication reliability for sidelink resources, a novel collision mitigation resource allocation scheme is proposed. The algorithm integrates the full-duplex detection capability of JCS with the resource sensing reservation process of the traditional SB-SPS scheme. This allows vehicles to dynamically optimize reservation times based on sensing channel information, effectively reducing consecutive packet collisions and enhancing the overall utilization efficiency of resources in the sidelink JCS system.The proposed CCM-SPS and traditional SB-SPS’s performance is theoretically analyzed in scenarios of a variety of vehicle densities and packet sizes. Through reinforcement learning, comprehensive optimization of resource utilization for sensing and communication is achieved.Comprehensive evaluations are performed using the Cramér–Rao Lower Bounds (CRLB), packet reception rate (PRR) and update delay (UD). The novel scheme shows comparative advantages in positioning accuracy, latency, and reliability performance indicators over a comparative scheme.

The remainder of the article is organized as follows. In Section 2, a review of recent literature is conducted, and in Section 3, a theoretical analysis is conducted on the performance of the sidelink JCS system. In Section 4, the consecutive collision problem is analyzed using a traditional resource allocation scheme. Then, in Section 5, the specific implementation of the improved resource allocation scheme is introduced, including full-duplex collision detection and a Q-learning based collision mitigation scheme. The extensive results are presented in Section 6, which analyzes the performance indicators in various scenarios. Section 7 presents the concluding remarks.

## 2. Related Works

Signal coexistence for achieving higher spectral efficiency has recently garnered significant attention, e.g., in [12,13,14,15,16]. Reference [12] proposes a system that combines collaborative communication technology with cognitive radio, which improves the overall performance of the system through cooperative spectrum-sharing networks. Reference [13] explored the coexistence of LAA and WIFI over unlicensed bands through a static contention window method. Their work provides practical solutions for achieving technological coexistence in existing network architectures. Reference [14] explores the advantages of integrating sensing and communication technologies (ISAC), including high spectral efficiency, low hardware costs, and improved system performance, as well as the potential for ISAC in future 6G applications. Reference [15] proposed a joint design framework for communication and sensing in small cellular networks, which optimizes the collaborative work of communication and sensing through waveform selection and resource allocation. Reference [16] addresses the practical challenges of residual hardware impairments (RHIs) and imperfect successive interference cancellation, deriving the superior performance of the ISAC framework compared to the sensing-communication coexistence (SCC) framework. It demonstrates that the integration of sensing and communication is a significant trend for future development.

In a highly dynamic V2X scenario, JCS systems can make network service adjustments more flexible and robust by simultaneously handling communication and sensing signals and dynamically coordinating [17]. Extensive research has focused on the JCS in vehicular networks, primarily concentrating on signal waveform design [18,19,20] and power management [21,22,23]. Firstly, extensive research on the design of signal waveforms provide the theoretical model foundation for the feasibility of JCS systems. Secondly, regarding interference and resource management, it can be found that existing research in JCS primarily focuses on large bandwidth millimeter-wave bands or Vehicle-to-Infrastructure (V2I) scenarios, which have certain limitations. The optimization of JCS system resource allocation is a challenge for V2V direct communication with limited resources and no base station assistance.

In Release 16, the 3rd Generation Partnership Project (3GPP) developed the NR-V2X and defined a new air interface PC5 as the sidelink, which allows for direct vehicle-to-vehicle communication to support various advanced use cases, covering four areas: platooning, extended sensing, remote driving, and autonomous driving [24]. In recent years, research on the sensing capabilities of NR-V2X sidelink signals has been increasingly explored. Studies on the theoretical foundation of CRLB for the sidelink sensing performance highlight the potential value for the resource allocation scheme of the JCS sidelink system. Reference [25] utilizes unused communication subcarriers of the 5G NR waveform as radar sensing subcarriers, aiming to minimize the CRLB of distance estimation by optimizing the amplitude of radar subcarriers, thereby improving sensing performance. Reference [26] derives the CRLB for distance/angle estimation accuracy based on sidelink OFDM signals, demonstrating the feasibility of sidelink JCS signals for near-field localization. References [5,27] derive the CRLB and mean squared error (MSE) for the sensing location of sidelink signals and analyze in detail the impact of communication physical layer parameters on sensing performance under interference conditions. Reference [28] compares the radar sensing performance of sidelink resource allocation SPS algorithms versus random allocation algorithms under interference, indicating that the JCS resource allocation mechanism is also a key factor affecting sidelink sensing performance. The above existing research contributions can be referenced to conduct a theoretical performance analysis of the sidelink JCS system in this paper.

According to the 3GPP standard [29], NR-V2X sidelink typically employs distributed resource allocation schemes based on the reservation mechanism, known as the SB-SPS algorithm [30], to obtain low latency communication performance. However, SB-SPS still faces consecutive resource collisions caused by resource allocation conflicts among multiple vehicles [31]. Recently, there have been many achievements in improving the SPS algorithm to alleviate the communication collisions [32,33,34]. Reference [32] improves the efficiency of resource utilization, enhances reliability, and reduces latency by designing a vehicle reuse distance during the SPS reselection process. Reference [33] develops an adaptive sidelink open-loop power control (AS-OLPC) algorithm to dynamically adjust transmission power, thereby improving communication reliability in complex urban environments. Reference [34] analyzes the effectiveness of introducing a re-evaluation mechanism at the MAC layer of NR-V2X to avoid collisions caused by packet re-transmissions, though the overall improvement is modest and requires more refined allocation strategies. However, in highly dynamic scenarios, there are uncertain complicated environment changes in the network that will intensify the packet collisions, so the above methods could not be adaptive to solve the communication performance degradation.

To overcome this problem, recent studies have increasingly focused on applying reinforcement learning techniques to optimize V2X resource allocation [35,36,37]. Existing resource allocation methods often rely on static models, which are unable to adapt in real-time to the dynamic traffic environment. In contrast, reinforcement learning can autonomously adjust, significantly improving resource utilization and system performance. Specifically, Reference [35] uses Q-learning to optimize SPS algorithm parameters, including reservation probability (RP) and reselection counter (RC), enhance packet reception rate (PRR) and reduce update delay (UD) in high-dynamic vehicular networks. Reference [36] proposes a deep reinforcement learning-based congestion control mechanism that optimizes channel busy rate (CBR) and age-of-information (AoI), showing significant improvements over traditional decentralized congestion control (DCC) algorithms. Reference [37] treats each vehicle as an independent agent and employs a multi-agent deep reinforcement learning (MARL) resource allocation algorithm, enabling vehicles to learn to select resource blocks and transmission power for periodic packet broadcasting. It is worthwhile to note from existing studies that the optimal RC parameter trained by RL can intelligently reduce consecutive collision probabilities in highly dynamic sidelink communication. Similarly, considering that RC optimization is crucial to sensing CRLB, this paper will introduce an RL approach to RC optimization in the JCS sidelink resource allocation to improve sensing performance.

Additionally, sidelink collision detection indicates that the dynamic change on channel states can be feasible through full-duplex technology. With advancements in self-interference cancellation (SIC) technology for in-band full-duplex communication [38], some studies have begun using full-duplex antennas to enhance resource allocation mechanisms. Reference [39] utilizes an in-band full-duplex transceiver and its collision detection capabilities to trigger SPS resource reselection, improving communication performance and analyzing the relationship between the collision detection threshold and vehicle density variations. Reference [40] also leverages collision detection capabilities brought about by full-duplex communication to adjust the probability of maintaining the same subchannel for transmission.

In short, most of the current resource allocation algorithm improvements only focus on improving communication performance, and there are few references on the joint optimization of sidelink JCS systems. Indeed, in practical sidelink scenarios with a limited frequency spectrum, the radar sensing accuracy can impact communication reliability due to the conflict in bandwidth requirements [28]. Due to the lack of base station coordination, a traditional SPS scheme may lead to serious consecutive packet collisions. To our knowledge, there is currently a lack of extensive research on joint optimization for sensing and communication in sidelink JCS systems. Therefore, this paper leverages pertinent research and proposes the CCM-SPS scheme to address the issue of consecutive collisions in sidelink JCS systems. This scheme utilizes full-duplex collision detection and Q-learning reinforcement learning methods, optimizing both the sensing range (CRLB) and ultra-reliable low latency communications (URLLCs) quality in vehicular JCS systems.

## 3. Theoretical Performance Analysis on Sidelink JCS System

Multiple vehicles simultaneously transmit data packets via full-duplex broadcasts and use echo signals to detect and locate passive targets in the surrounding environment, obtaining sensing information such as target distance, relative speed, and the signal-to-noise ratio (SNR). In this scenario, multiple vehicles send sidelink signals, such as a cooperative awareness message (CAM) or collective perception message (CPM) [41]. It is initially assumed that all vehicles are equipped with in-band full-duplex transceivers with perfect SIC, enabling the use of echoes to detect channel quality [42].

Assume that during time slot t∈T, vehicle $i\in I_{t}$ generates a data packet and begins broadcasting the OFDM symbols $s_{m}\left(t\right)$. In this context, the symbol for the *n*-th subcarrier of the *m*-th symbol is denoted as $x_{n,m}$, with each symbol power being $P_{n,m}$. The symbol duration is defined as $T_{sym}=T+T_{cp}$, where $T_{cp}$ represents the cyclic prefix duration and T=1/Δf, where Δf is subcarrier space. The representation in the complex baseband is as follows:(1)$$s_{m}\left(t\right)=\sum_{n=0}^{N-1}\sqrt{P_{n,m}}x_{n,m}e^{j2\pi n\Delta ft}\mathrm{rect}\left(\frac{t-mT_{\mathrm{sym}}}{T_{\mathrm{sym}\;}}\right)$$

The sidelink JCS data packet consists of *N* subcarriers and *M* OFDM symbols. Assume that all the symbols $x_{n,m}$ in the packet form an N×M matrix $X=\left\{x_{n,m}\right\}\in C^{N\times M}$, where each column represents an OFDM symbol and each row represents a subcarrier. Assume that the power of each symbol on each subcarrier is the same and normalized, i.e., $P_{n,m}=P_{avg}$. Therefore, for all *N* and *M*, the transmission power of each vehicle is given by $P_{T}=N\cdot P_{avg}$.

Sidelink data packets are not only used for communication transmission but also for echo sensing. Assume the target vehicle is *d* away from the transmitting vehicle and the relative velocity is *v*. The complex baseband representation of the received echo signal for the *n*-th subcarrier of the *m*-th symbol is given by:(2)$$y_{m,n}\approx \alpha \;e^{j2\pi mT_{\mathrm{sym}}\nu }e^{-j2\pi n\Delta f\tau }x_{m,n}$$

Among them, α is the channel coefficient. By receiving the echoes at the transmitting receiver, estimates of ν and τ are obtained as $\hat{\nu }$ and $\hat{\tau }$. The parameters ν and τ represent the Doppler shift and delay of the echo signal, respectively. The distance estimate $\hat{d}$ is calculated from $\hat{\tau }$ as:(3)$$\hat{d}=\frac{\hat{\tau }}{2c}$$
where *c* represents the speed of light. The carrier frequency is $f_{x}$. The relative speed estimate $\hat{v}$ is calculated from $\hat{\nu }$ as:(4)$$\hat{v}=\frac{\hat{\nu }c}{2f_{x}}$$

$P_{R}$ is the received power of the full duplex, given by:(5)$$P_{R}=P_{T}\frac{G^{2}c^{2}\sigma }{{\left(4\pi \right)}^{3}f_{c}^{2}d^{4}}$$

Among them, $P_{T}$ represents the transmit power, and *G* represents the transmit/receive antenna gain. The noise power is represented as $P_{n}=k_{B}T_{0}FW$, where $k_{B}$ is the Boltzmann constant, $T_{0}=290$ is the reference temperature, *F* is the noise figure of the full-duplex radar receiver, and the bandwidth of the sidelink transmission signal is W=NΔf.

In practice, during the transmission slot, interference occurs when other vehicles transmit using the same frequency resources, which affects radar sensing performance. The calculation formula is:(6)$$I_{t,i}=\sum_{k\in I_{t},k\neq i}\eta_{t,ki}\frac{h_{t,ki}P_{T}}{L(d_{t,ki})}$$

In (6), $L\left(d_{t,ki}\right)$ represents the path loss at distance $d_{t,ki}$ between the interfering vehicle *k* and the transmitting vehicle *i* at time slot *t*, and the parameter $h_{t,ki}$ represents the large-scale fading at time slot *t*. Additionally, $\eta_{t,ki}$ is a coefficient that takes values of 0 or 1, indicating the absence or presence of interference in that time slot, respectively.

Therefore, considering Gaussian white noise and interference from other vehicles, the signal-to-interference-plus-noise ratio (SINR) of the reflected echo signal from a target located at a distance *d* from the transmitter is calculated as follows:(7)$${\mathrm{SINR}}_{t,i}=\frac{P_{R}}{P_{n}+I_{t,i}}$$

According to [43,44], the radar sensing performance of NR-V2X signals with OFDM waveforms is represented by the Cramér–Rao Lower Bound (CRLB) on distance estimation variances. The CRLB represents the best performance that can be achieved for unbiased estimation of these parameters. The CRLB for distance estimation for the transmitting/sensing vehicle *i* at time slot *t* is given by:(8)$${\mathrm{CRLB}}_{t,i}^{\left(\hat{d}\right)}=\frac{3c^{2}}{{\mathrm{SINR}}_{t,i}8\pi^{2}\Delta f^{2}MN\left(N^{2}-1\right)}$$

The CRLB clearly describes the most optimistic performance achievable and serves as a benchmark for characterizing sensing accuracy. Practically, the signal processing methods cannot achieve performance below the theoretical CRLB.

Further, based on the preceding analysis, the impact of sidelink bandwidth resources on sensing performance can be analyzed. According to (8), the CRLB for distance estimation decreases rapidly with the cube of the bandwidth. Therefore, increasing the subcarrier spacing (Δf) or the number of subcarriers (*N*) is advantageous for distance estimation.

Simultaneously, according to Shannon’s theorem, increasing bandwidth benefits the data rate in vehicular networks. However, in a multi-user distributed resource allocation system with limited resources, indiscriminately increasing *N* to enhance sensing and communication performance may exacerbate consecutive packet collisions within the existing sidelink resource allocation strategy. This interference significantly impacts the performance of the JCS system and reduces resource utilization efficiency [28]. The subsequent sections will provide a detailed analysis of optimizing the allocation scheme to mitigate collisions.

## 4. Consecutive Collision Problem Analysis of the JCS Sidelink Resource Allocation

### 4.1. Principle of the SB-SPS Resource Allocation Scheme

NR-V2X supports Mode 2 sidelink communication and employs the sensing-based semi-persistent scheduling (SB-SPS) scheme [45]. Initially, the SB-SPS scheme is designed to support periodic safety messages, utilizing sensing windows and a resource reservation mechanism to reduce end-to-end latency. The fundamental working principle of SB-SPS is illustrated in Figure 2, with specific steps as follows:

In the sensing window, vehicles measure the reference signal received power (RSRP) of a physical resource block (PRB), continuously generating a list of available resources $L_{a}$. This list includes time-frequency resources with RSRP values below the threshold $P_{th}$. Once the number of resources in $L_{a}$ is less than X% of the total resources, $P_{th}$ will increase by 3 dB to increase the number of $L_{a}$. The X% threshold can be set to 20%, 35%, or 50% depending on the configuration and service priorities. Subsequently, during the resource selection window, a reselection counter (RC) is employed to manage the use of reserved resources, with resource reselection occurring only when the RC reaches 0. This process involves selecting a new resource with probability $\left(1-P_{k}\right)$ or continuing to use the previously reserved resource with probability $P_{k}$, where $P_{k}$ ranges from 0 to 0.8. Once a resource is selected, continuous transmission occurs in the same resource block, with the number of transmissions determined by the value of RC. RC is randomly chosen within a range between 5 and 15 and decremented by 1 after each transmission until RC reaches 0, at which point the next selection process is triggered. Equation (Equation 9) demonstrates the ΔRC, which is the RC decreasing step size of the SB-SPS:(9)$$\Delta RC=RC^{\prime }-RC=1$$

Due to the distributed resource allocation characteristics of the sidelink, it is impossible to obtain complete channel state information, which means that it cannot be guaranteed that all vehicles select idle resources. In addition, there is partial overlap in the candidate resource pools between neighboring vehicles, resulting in packet collisions. This issue is exacerbated in dynamic scenarios with high-density and large packet-size services. Furthermore, the sidelink echo signals are also subject to interference from consecutive packet collisions, which degrade the sensing performance.

### 4.2. Markov Chain Model of SB-SPS

In this section, a Markov chain analytical model [46] is presented, as shown in Figure 3. At any slot *t*, the corresponding RC(t)∈[0,15]. If RC(t)=0, it is randomly re-initialized s.t. RC(t+1)∈[5,15]. Thus, the probability that(10)$$\mathrm{Pr}\left\{RC\left(t+1\right)=i|RC\left(t\right)=0\right\}=\frac{1}{11}.$$

Denote $\pi_{i}$ as the probability that RC(t)=i,0≤i≤15. According to Figure 3, $\pi_{i}$ satisfies:(11)$$\pi_{i}=\left\{\begin{array}{cc} \pi_{0} & \mathrm{for}\;0\leq i\leq 4,\\ \frac{1}{11}\pi_{0}+\pi_{i+1} & \mathrm{for}\;5\leq i\leq 14,\\ \frac{1}{11}\pi_{0} & \mathrm{for}\;i=15. \end{array}\right.$$

Using the normalization condition $\sum_{i=0}^{15}\pi_{i}=1$, and solving (10), we obtain $\pi_{0}=\frac{1}{11}$.

Since access collisions are caused by resource reselection, we first define the probability of a reselection resource collision. A collision will occur when multiple vehicles reselect the same resources within overlapping selection windows. Assume that at time *t*, vehicle UE0 is in the state RC=0, and it performs a reselection during the selection window [t,t+RRI]. During this time, other UEs also engage in reselection. UEs transition to RC=0 with probability $\pi_{0}$ and reselect with probability $1-P_{k}$, moving to a new state. Since the RC states of each UE are independent, if *n* out of $N_{\mathrm{UE}}$ UEs are involved in reselection, the probability that other vehicles also trigger reselection when UE0 triggers reselection is given by:(12)$$\begin{array}{cc} & P_{s}\left(n\right)=\mathrm{Pr}\left\{n\;RC_{UEs}=0,\;\mathrm{Reselect}|RC_{UE0}=0,\;\mathrm{Reselect}\right\}\\ & =\left(\begin{array}{c} N_{UE}\\ n \end{array}\right){\left(\pi_{0}\left(1-P_{k}\right)\right)}^{n}{\left(1-\pi_{0}\left(1-P_{k}\right)\right)}^{N_{UE}-n} \end{array}$$
where *n* represents the number of vehicles simultaneously reselecting within UE0’s reselection window. Access collision may occur if at least one other vehicle selects the same available PRB as UE0.

We define the collision involving *n* UEs reselecting within the overlapping selection window as an *n*-fold collision, given by the following formula:(13)$$\begin{array}{cc} & P_{r}\left(n\right)=\mathrm{Pr}\left\{n-\mathrm{fold}\;\mathrm{Collision}|n\;RC_{UEs}=0,\;\mathrm{Reselect}\right\}\\ & =1-{\left(1-\frac{1}{\bar{N_{a}}}\right)}^{n} \end{array}$$
where $\bar{N_{a}}$ represents the average number of available PRBs within the selection window. This number is influenced by the vehicle density and the packet size. Higher vehicle density and larger packet sizes result in fewer available resources.

Thus, when UE0 performs reselection within the selection window, the access collision probability can be obtained as follows:(14)$$\begin{array}{cc} & P_{c}=\mathrm{Pr}\left\{\mathrm{Collision}|RC_{UE0}=0,\;\mathrm{Reselect}\right\}\\ & =\sum_{n=1}^{N_{UE}}P_{r}\left(n\right)P_{s}\left(n\right) \end{array}$$

In SB-SPS, the same resources would be continuously colliding after an access collision. The collision will last for at least $\min[RC_{\mathrm{UE}0},RC_{\mathrm{UE}1}]$ times between two UE0 and UE1. Therefore, it is useful to adjust ΔRC, which is the RC decreasing step size to mitigate consecutive collisions. Increasing the ΔRC can reduce consecutive collisions. However, an overly aggressive increase in ΔRC will increase the number of vehicles entering the selection window simultaneously, thereby increasing the probability of access collision, according to (14). Therefore, it is necessary to optimize the ΔRC.

## 5. Q-Learning-Based CCM-SPS Resource Allocation Scheme Proposed for JCS Sidelink

Details of our proposed CCM-SPS scheme to improve the sensing and resource reservation process will be elaborated below.

### 5.1. Collision Detection Mechanism

The full-duplex (FD) transceiver calculates the echo power as a condition for collision detection. The total received power of the FD receiver can be expressed as:(15)$$P_{r}^{FD}=P_{R}+P_{n}+I_{t,i}$$
where $P_{R}$ represents the power of the reflected echo signal. Assume only the closest signal to the transmission vehicle is considered. Since the CAM signal contains distance information, $P_{R}$ can be calculated through path loss using the distance of CAM messages. $P_{n}$ is the Gaussian white noise power. $I_{t,i}$ represents the interference of other transmitting vehicles to VUE *i* in the same time slot *t* according to (6).

By calculating the difference between the received power $P_{r}^{FD}$ and the echo power $P_{R}$ from the nearest vehicle, while also accounting for the noise power $P_{n}$, the interference power strength $I_{t,i}$ can be effectively estimated.

Then, the collision detection rules for collisions in the JCS sidelink are given as follows:(16)$$\left\{\begin{array}{cc} P_{r}^{FD}-P_{R}-P_{n}>P_{\Delta }, & COL=1\\ P_{r}^{FD}-P_{R}-P_{n}<P_{\Delta }, & COL=0 \end{array}\right.$$

The threshold is used to determine whether the interference impact exceeds the critical power. If COL=0, this indicates no resource collision; if COL=1, this indicates a resource collision.

### 5.2. Q-Learning-Based CCM-SPS Scheme for JCS Sidelink

After detecting a resource collision, a consecutive collision elimination mechanism based on Q-learning is proposed to mitigate consecutive resource collisions. Vehicles interact with the environment in real-time and intelligently determine the optimal actions given the current state.

#### 5.2.1. Vehicular Agent Based on the Reinforce Learning Model

In a typical reinforcement learning framework [47], as illustrated in Figure 4, the agent achieves its learning objectives through an iterative process of receiving rewards from the environment. Intelligent agents improve their strategies by exchanging rewards with the environment.

In the proposed sidelink JCS system, each vehicle updates its current state after transmitting data packets and detecting collisions. The current state includes the collision status and the current RC value. For vehicle *i*, the state $s_{i}\in S$ can be described as follows:(17)$$s_{i}=\left(COL_{i},RC_{i}\right)$$
where ${\mathrm{COL}}_{i}$ indicates whether the vehicle *i* experiences a resource collision when transmitting a data packet. ${\mathrm{RC}}_{i}$ represents the current value of the reselection counter, with a range of [0, 15]. Therefore, the state set consists of a discrete state space.

The traditional SB-SPS scheme reduces the RC by 1 after each packet transmission. To regulate the decreased efficiency of RC, we propose a set of a discrete action space that represents the selection of actions based on acquired state information. This enables resource collision vehicles to adjust the RC decreasing step size (ΔRC).(18)$$A=\{a_{i}\in \Delta RC\}$$
where ΔRC represents the value of the RC that the vehicle *i* has to reduce after each transmission. The vehicle will dynamically select an action in the action space of [0, 15] based on the observed state. When the RC−ΔRC<0, the vehicle enters the reselection process and reselects a new resource.

After a vehicle executes an action, its state will transition, and it will learn the instantaneous reward *R* from the environment.(19)$$R=\left\{\begin{array}{cc} \left(\exp\left(-N_{col}\right)-1\right)/log_{10}\left(RC+1\right), & \mathrm{COL}=1\\ 1, & \mathrm{COL}=0 \end{array}\right.$$
where $N_{col}$ represents the times of a consecutive resource collision for the vehicle. When the collision does not occur, the reward is 1. When a collision occurs, the greater the $N_{col}$ is, the greater the penalty is, ensuring that the collision state can be removed as quickly as possible. In addition, the current RC value is introduced as a constraint. The smaller the RC, the greater the penalty. When different RC vehicles collide in the same resources, the probability of the vehicle with a small RC entering the reselection stage increases, while the vehicle with a large RC can continue to transmit using the current resources. This mechanism can effectively reduce the number of vehicles in the reselection stage simultaneously, thus helping to maintain the stability of the system.

In Q-learning, Q value Q(s,a) was calculated and updated using a reward model to evaluate the state-action mapping policy under the state action pair (s,a). The Q value is updated by the Bellman equation, as shown below:(20)$$\begin{array}{cc} Q(s,a) & \leftarrow Q(s,a)+\alpha [R\\ & +\eta \max_{s\in S}Q\left(s^{\prime },a\right)-Q\left(s,a\right)] \end{array}$$

When vehicles optimize the search for the optimal action, they need to balance the exploitation and exploration of learned knowledge to ensure that each action has a possibility of being selected.(21)$$\left.a=\left\{\begin{array}{cc} \arg\max_{a\in A}Q\left(s,a\right), & \mathrm{with}\;\mathrm{prob}.\;1-\varepsilon \\ \mathrm{random}, & \mathrm{with}\;\mathrm{prob}.\;\varepsilon \end{array}\right.\right.$$

This paper adopts an ϵ-greedy strategy to balance the exploitation–exploration process. During the training process, the initial ϵ is set to 0.9 and gradually decreased to 0.1. The training lasted for 100 s, consisting of 1000 iterations. The initial learning rate was set to 0.01, and the discount factor was set to 0.9.

#### 5.2.2. Algorithm Flow and Pseudo-Code

Based on previous analysis, we proposed the CCM-SPS scheme to reduce the times of consecutive collisions. Specifically, during transmission, the full-duplex (FD) echo detection capability is used to sense the channel state. If a collision is detected, an ΔRC is selected based on the current state. This means that when a collision is detected, the scheme decreases the times of consecutive collision and restricts the number of vehicles entering the reselection process simultaneously, as shown in Figure 5.

The pseudocode of its algorithm is shown below (Algorithm 1):
**Algorithm 1:** Pseudo-code of the proposed CCM-SPS**Input:** Vehicle density, Packets occupy bandwidth**Output:** 
Qtable   1:Initialize the parameters such as learning rate α and discount factor η.   2:Initialize the vehicles’ states, actions, and Qtable   3:**loop**   4:    Begin the new packet transmission   5:    Observe transmitted vehicle $s_{i}$   6:    **if** collision occurred, COL=1 **then**   7:        the number of consecutive resource collisions        $N_{col}=N_{col}+1$   8:        Vehicle *i* obtain the negative reward:        $\left(\exp\left(-N_{col}\right)-1\right)/log\left(RC+1\right)$   9:    **else** 10:        the number of consecutive resource collisions        $N_{col}=0$ 11:        Vehicle *i* obtain the positive reward 1 12:    **end if** 13:    Vehicle update the Q(s,a) 14:    Vehicle update the probability ϵ according to simulation time 15:    **if** exploration **then** 16:        Vehicle randomly select an action α 17:    **else** 18:        Vehicle select the optimal action $\alpha_{opt}$ with $Q_{max}$ 19:    **end if** 20: **end loop**

## 6. JCS Sidelink Performance Evaluation Using CCM-SPS

In this section, key performance metrics such as CRLB, PRR, and UD were evaluated under varying vehicle density and packet size scenarios, and comprehensive experimental discussions were conducted. Among them, CRLB is a sensing performance metric that affects target detection accuracy, thereby influencing autonomous driving decision-making; PRR is a communication reliability metric that impacts communication quality; and UD is a latency metric that affects the real-time performance of intelligent transportation systems.

### 6.1. Simulation Setup

The main settings are reported in Table 1 and discussed hereafter.

In this section, we simulate the performance of a JCS sidelink system using a system-level simulator. In particular, vehicles perform SINR evaluation for collision detection and CRLB calculation after each transmission.

Scenario. We simulated a three-lane bidirectional highway scenario, with vehicle density varying between 50, 150, and 250, depending on the setting. The average speed of vehicles was 70 km/h, and the STD of vehicle speed was 7 km/h. The radar cross-section of the vehicle is 10 dBsm.

Power settings and channel model. The channel model is WINNER + B1, with a fixed available bandwidth of 40 MHz and a center frequency of 5.9 GHz, to simulate a V2V channel scenario. The transmission power of the vehicle is 23 dBm, assuming that the gain of both the transmitting and receiving antennas is 3 dBi and the noise gap is 6 dB.

Physical layer and data traffic. In terms of physical layer settings, we used fixed SCS and MCS sizes. In terms of data traffic settings, we simulated V2V periodic message transmission with a period of 100 ms and support packet sizes of 350 or 1000 bytes to represent CAM and CPM business types, respectively. The packet size, under fixed MCS and SCS configurations, will affect the bandwidth occupied by vehicle transmission packets, as shown in Table 2.

### 6.2. JCS Sidelike Performances with Dynamic Vehicle Density

Firstly, the JCS performances of the proposed CCM-SPS have been evaluated, compared with a benchmark scheme in the case of various vehicle densities. The benchmark scheme, FD-enhanced, is proposed in [39]. It detects resource collisions using full-duplex during transmission and then uses aggressive resource reselection by setting RC to 0 for all vehicles for which access collision occurs. It can quickly break off consecutive collisions to somehow improve communication performance. The proposed CCM-SPS scheme employs Q-learning to optimize the resource reselection process in addition to the FD detection in the resource reservation process.

Figure 6 illustrates the empirical CDF of the root CRLB for a range using SB-SPS, FD-enhanced and CCM-SPS in various density scenarios. As the vehicle density increases, the sensing performance of the root CRLB range decreases. And the FD-enhanced significantly improves performance at medium and low vehicle densities, but the improvement is not as obvious at high vehicle densities. However, the proposed CCM-SPS not only further enhances performance in medium to low-density but also effectively improves performance in high-density scenarios. Figure 7 presents 95% range root CRLB for different algorithms at varying densities. The bar graph provides a more visual representation showing the superior performance of the proposed CCM-SPS across all density scenarios compared to the other two schemes.

Similar results also appear in discussions on communication performance. Figure 8 illustrates PRR over a distance using SB-SPS, FD-enhanced and CCM-SPS in different density scenarios. As the vehicle density increases, the PRR of different schemes all show a downward trend. In mid- and low-density scenarios, the FD-enhanced scheme shows some improvements in PRR within a 200 m communication range. However, in high-density scenarios, the improvements are not significant. By examining the maximum distance where PRR exceeds 0.95 under three different schemes at varying densities, as shown in Figure 9, the advantages of the CCM-SPS scheme become more obvious. Compared to the FD-enhanced scheme, the proposed algorithm can further enhance communication metrics effectively across different density scenarios, especially in high vehicle density. The maximum communication ranges in low-, mid-, and high-density scenarios are increased by 22.2%, 30%, and 50%, respectively. This analysis indicates that the CCM-SPS scheme demonstrates robustness and effectiveness in various traffic density scenarios.

In short, as vehicle density rises, available resources in the pool decrease, increasing the probability of access collisions and resulting in consecutive resource collisions, leading to JCS performance degradation. The FD-enhanced method employs full-duplex to detect collisions and achieve early termination of consecutive collisions. However, an overly aggressive reselection mechanism leads to an excessive number of vehicles entering the reselection stage in high-density scenarios. According to (13), an increase in the number of vehicles in the reselection process within the same time slot raises the probability of new access collisions. The advantages of the FD-enhanced scheme over traditional SB-SPS are more pronounced at mid and low densities but diminish at high densities. However, the CCM-SPS scheme outperforms FD-enhanced in a variety of density scenarios. It can be demonstrated that the Q-learning-based scheme significantly enhances performance, particularly in high-density environments. This is due to the reward function defined in the proposed Q-learning algorithm, i.e., vehicles that do not undergo resource collisions will receive rewards and strive to maintain the current state as much as possible, ensuring the stability of the resource pool. Vehicles that experience resource collisions are penalized based on the current RC and the times of consecutive collision, which encourages vehicles to learn the best strategy to minimize consecutive collisions and avoid new collisions.

The results demonstrate that the proposed CCM-SPS scheme outperforms the comparison methods in terms of CRLB and PRR across various density scenarios. These advantages have strong application potential in fields of high-accuracy sensing. For example, in high-density environments, it maintains a CRLB accuracy of 0.1 m and a reliable communication range of 50 m, ensuring both accurate target sensing and reliable communication, which are critical for autonomous driving decision-making.

### 6.3. JCS Sidelike Performances with Dynamic Packet Sizes

#### 6.3.1. Conflicting Impacts of Packet Sizes on JCS Performance Metrics

Secondly, in this section, we investigate the conflicting impacts of packet sizes on JCS performance metrics in terms of root CRLB range and PRR in different vehicle density scenarios by using a traditional SB-SPS scheme. Then, we evaluate the optimization performance results of a Q-learning-based CCM-SPS scheme, including sensing and communication metrics in different vehicle density scenarios.

Figure 10 illustrates the empirical CDF of root CRLB range estimation with different packet sizes and vehicle densities. With a given pack size, the empirical CDF of the root CRLB Range decreases rapidly with an increase in vehicle density. As the packet size increases, the CDF curve of the root CRLB Range shifts toward the left with a small value of CRLB. This result indicates the enhanced sensing accuracy performance.

Figure 11 illustrates the communication reliability metric PRR vs. distance under different packet sizes and vehicle densities. With a given packet size, the communication reliability PRR decreases with the transmission distance. As the packet size or the vehicle density increases, the PRR curves decline fast.

Based on the above discussion of the results of both Figure 10 and Figure 11 of the JCS system with the traditional SB-SPS scheme, it can be seen that with increasing vehicle density, both sensing and communication performances decrease. This is because the worse channel quality of the sidelink in a high-density vehicle network yields more consecutive collisions over the shared resource pool with traditional SB-SPS. Moreover, it also can be seen that with increasing packet size, the sensing accuracy increases with a smaller root CRLB range value, while the communication reliability declines with a smaller PRR value. In the case of a big packet size, the required bandwidth for transmission increases when more subcarriers are allocated. An echo radio with a large subcarrier number N can produce a small CRLB value range, according to (8). At the same time, there is a high probability that different vehicles occupy the same spectrum resource, resulting in serious consecutive collisions due to the resource competition over the sidelink, which would deny successful access and produce a small PRR.

In short, the traditional sidelink resource allocation scheme is less effective for a sidelink JCS system to support various services in highly dense dynamic vehicular networks. In order to overcome the conflicting requirement on the spectrum resource allocation between sensing and communication, this paper has proposed a novel CCM-SPS scheme to realize an effective sidelink resource allocation for enhancing JCS performances. Next, in the following section, JCS performances will be evaluated.

#### 6.3.2. Optimization Performance Evaluation of Q-Learning-Based CCM-SPS

The proposed CCM-SPS can optimize JCS performance metrics by introducing the Q-learning method in order to control the repetition times of the reserved resources by adjusting a decreasing RC step size and suppressing the consecutive collisions probability.

Figure 12 illustrates the CCM-SPS’s range sensing performance evaluation on an empirical CDF of root CRLB with different pack sizes in the case of density = 50, 150, 250 veh/km. The performances of traditional SB-SPS with the same configuration are also given as a comparison. As for the CDF curve of the root CRLB Range, CCM-SPS’s performance curve increases faster than traditional SB-SPS at a given packet size and vehicle density, giving a high probability of a small CRLB value. Meanwhile, with an increase in packet size, the larger the packet size, the more CRLBs. As the vehicle density increases, CCM-SPS has the advantage of being able to maintain a smaller CRLB value than SB-SPS.

On the other hand, Figure 13 shows PRR vs. distance performance of CCM-SPS with different pack sizes with different densities. Figure 14 indicates CCM-SPS’s communication performance evaluation on empirical CDF of update delay with different pack sizes with different densities. The performances of traditional SB-SPS with the same configuration are also given as a comparison. Figure 13 and Figure 14 provide the communication performance results.

It can be seen that, at a given vehicle density, CCM-SPS can obtain a higher PRR than SB-SPS. With the packet size increase, the PRR of CCM-SPS decreases slower than SB-SPS. As the vehicle density becomes large, CCM-SPS can hold a relatively high PRR and SB-SPS a low PRR even in the low density scenario Similarly, with a given vehicle density, CCM-SPS can achieve an obvious deduction on update delay compared with SB-SPS. As vehicle density becomes dense, the update delay increases in both schemes, but CCM-SPS can maintain a high probability of a small update delay value. Therefore, the proposed CCM-SPS can achieve optimal communication qualities, such as high reliability and low latency, better than the traditional scheme.

In view of the above discussion results, it is significant that CCM-SPS can fulfill comprehensive performance enhancements on both sensing and communication without affecting the cost of each other. CCM-SPS makes resource reservation feasible for sidelink JCS access after FD detects the dynamic SINR from the echo signal. According to reward function (19) related to RC and $N_{col}$, the JCS vehicle agent can learn from the dynamic network environment and feed back a corresponding reward in order to optimize the ΔRC selection actions in the reservation process through the Q-learning approach. As a result, the effective resource allocation for both sensing and communication can simultaneously be realized by using CCM-SPS. Therefore, the CCM-SPS algorithm is capable of supporting dynamic data traffic service scenarios such as the V2X network in intelligent transportation systems.

## 7. Conclusions

This paper proposes a resource allocation scheme in a sidelink JCS system, named consecutive collision mitigation semi-persistent scheduling (CCM-SPS). By employing collision detection referring to the echo power threshold and Q-learning to train the RC decreasing step size, this scheme can effectively suppress the consecutive collision probability. Compared with traditional SB-SPS and the FD-enhanced scheme, CCM-SPS can achieve both superior sensing and communication performance even in high-density vehicle scenarios. Furthermore, CCM-SPS can support services with large packet sizes and achieve accurate sensing, and the cost of communication reliability is smaller as the distance increases. It is particularly meaningful for CCM-SPS from the perspective of enabling sidelinks to support sensing and communication collaboration in 6G networks. In future work, there are interesting topics to be studied, such as practical full-duplex impacts from interference and cross-layer optimization. In addition to the V2X network studied in this paper, there is still room to explore the CCM-SPS scheme to be used in various JCS applications, such as the AIOT network. Additionally, the integration of edge computing with the CCM-SPS scheme can further enhance the performance of the Sidelink JCS system to support rich and broad JCS application tasks.

## Figures and Tables

**Figure 1 sensors-25-00302-f001:**
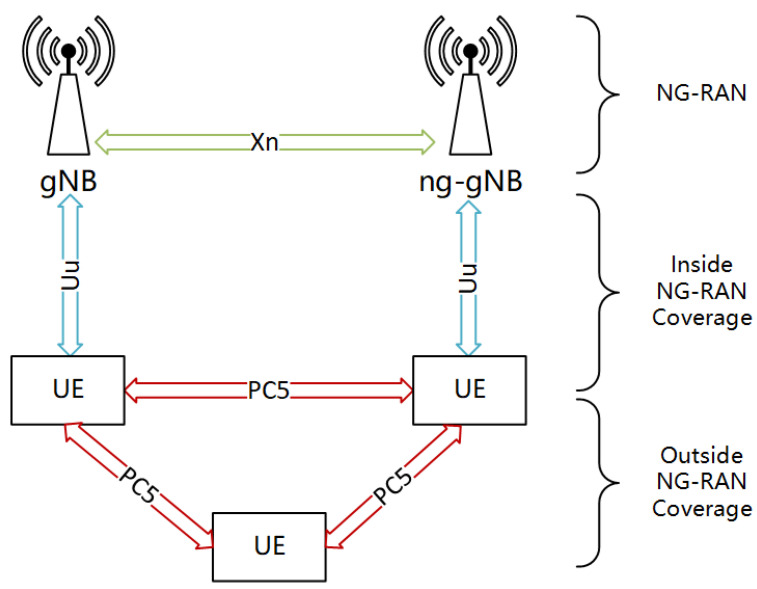
NG-RAN architecture supporting the PC5 interface.

**Figure 2 sensors-25-00302-f002:**
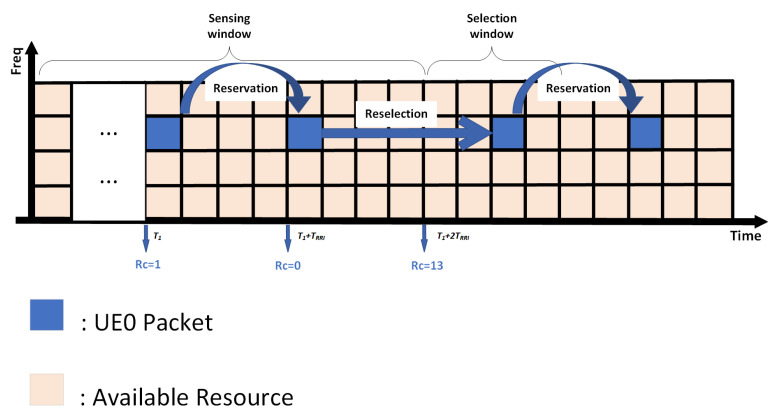
Process flow of sensing-based semi-persistent scheduling (SB-SPS).

**Figure 3 sensors-25-00302-f003:**
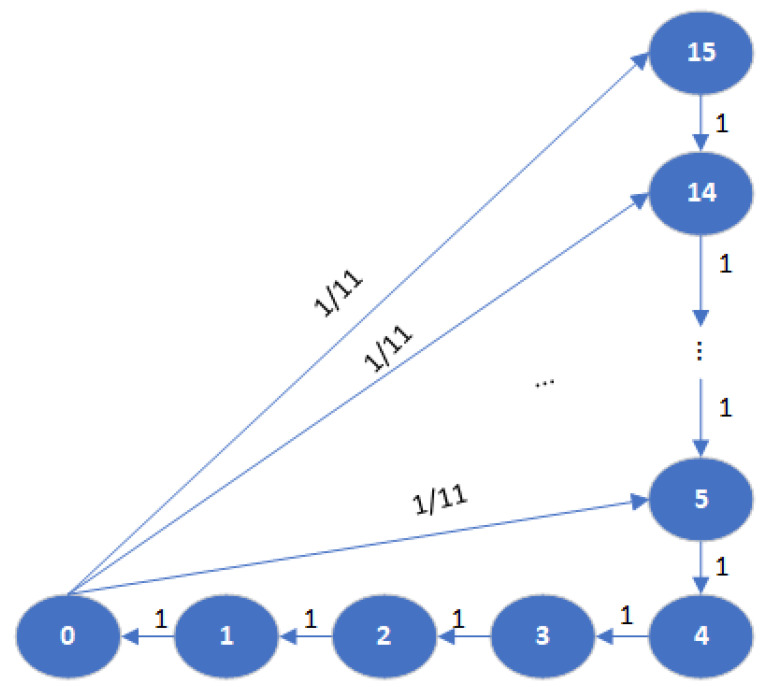
Markov chain for state transition of SPS.

**Figure 4 sensors-25-00302-f004:**
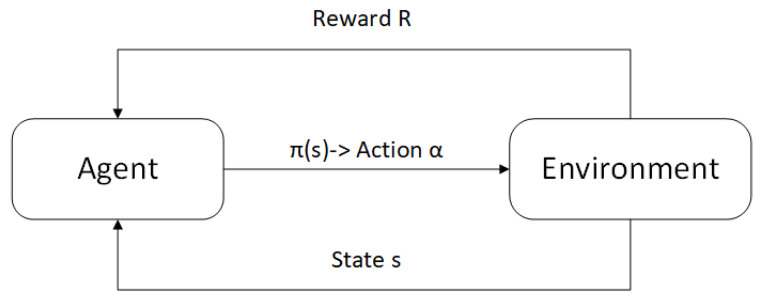
Reinforcement learning framework.

**Figure 5 sensors-25-00302-f005:**
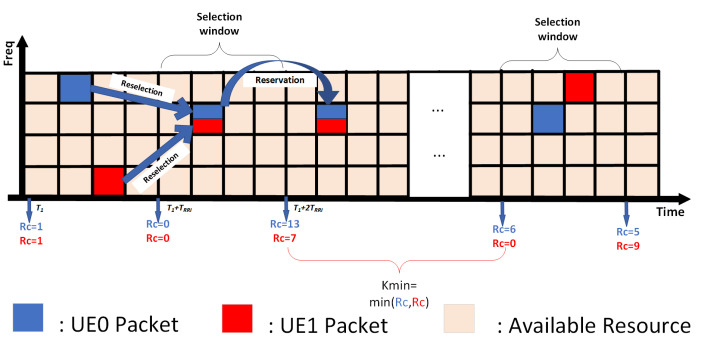
CCM-SPS accelerate reselection.

**Figure 6 sensors-25-00302-f006:**
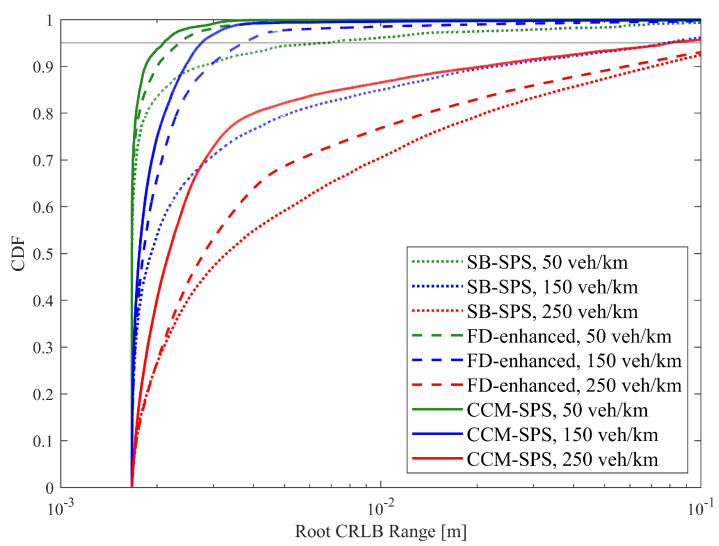
Empirical CDF of the root CRLB for a range using SB-SPS, FD-enhanced and CCM-SPS.

**Figure 7 sensors-25-00302-f007:**
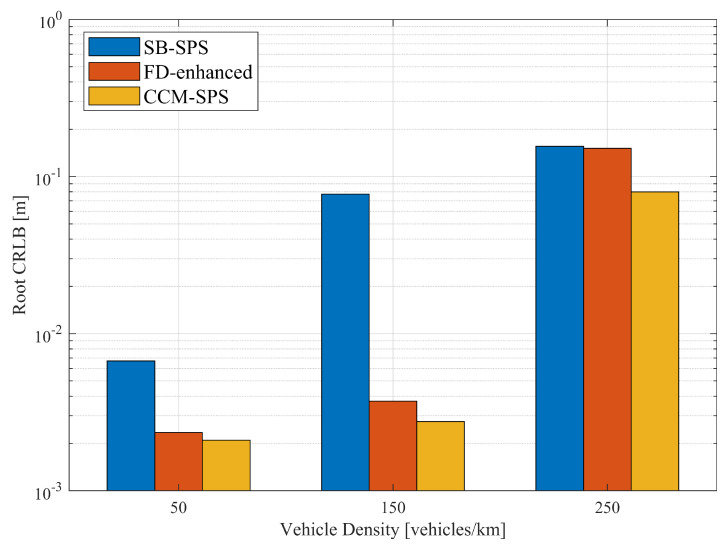
Bar graph of root CRLB (at CCDF = 95-percentile) for a range using different schemes with varying vehicle densities.

**Figure 8 sensors-25-00302-f008:**
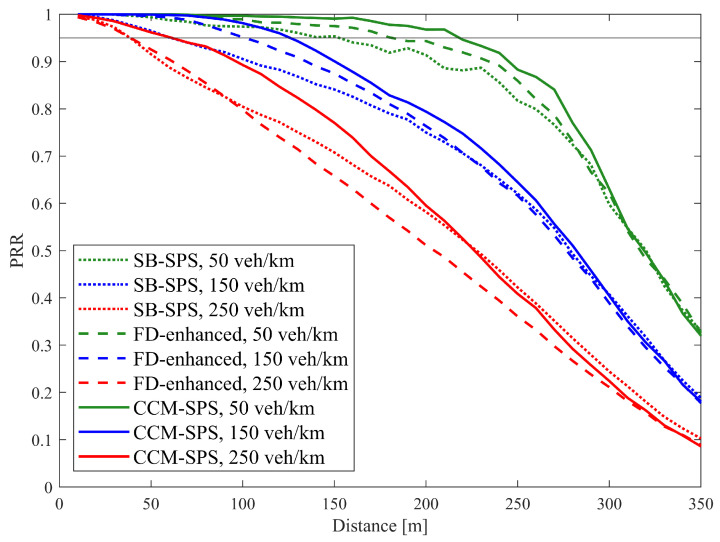
PRR over distance using SB-SPS, FD-enhanced and CCM-SPS.

**Figure 9 sensors-25-00302-f009:**
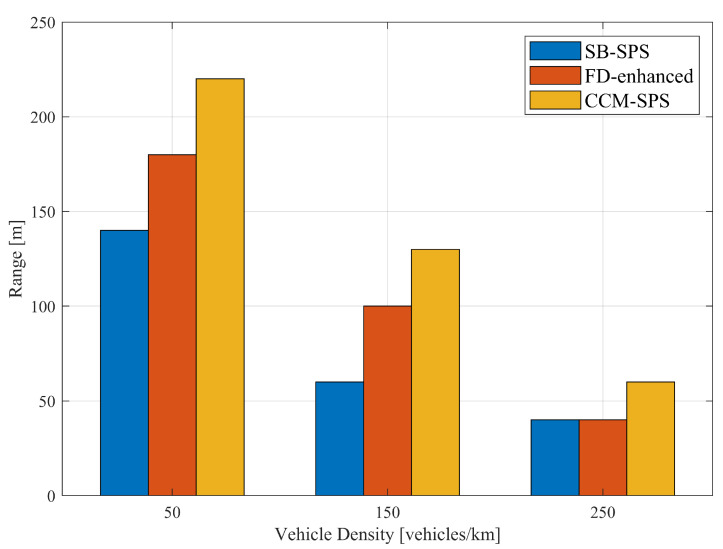
The maximum distance allowing PRR larger than 0.95 is evaluated using conventional SB-SPS, FD-enhanced methods and CCM-SPS.

**Figure 10 sensors-25-00302-f010:**
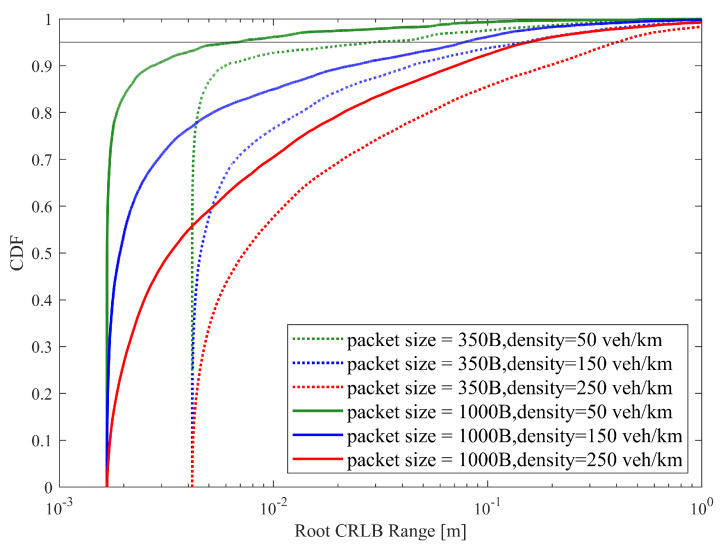
Empirical CDF of root CRLB for range estimation.

**Figure 11 sensors-25-00302-f011:**
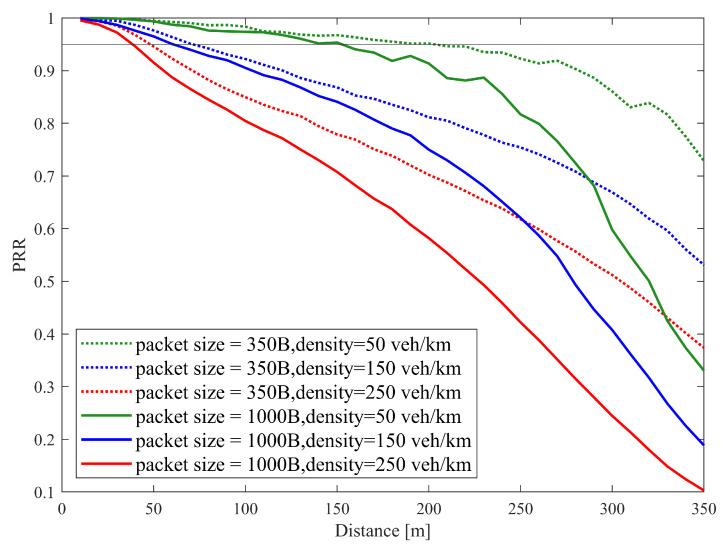
PRR vs. distance performance of SB-SPS with different pack sizes in case of density = 50, 150, 250 veh/km.

**Figure 12 sensors-25-00302-f012:**
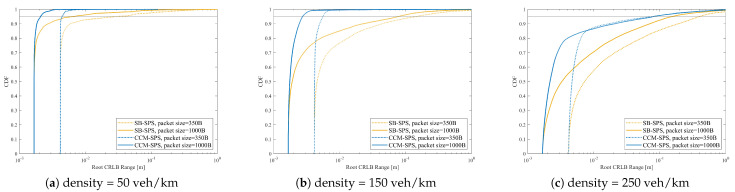
CCM-SPS’s range sensing performance evaluation on empirical CDF of root CRLB with different pack sizes in the case of density = 50, 150, 250 veh/km.

**Figure 13 sensors-25-00302-f013:**
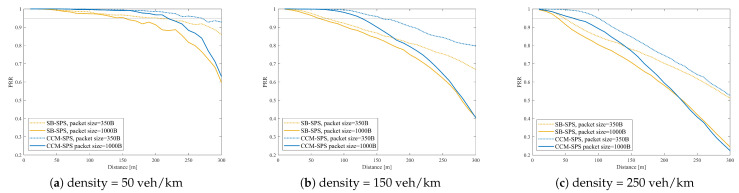
PRR vs. distance performance of CCM-SPS with different pack sizes in case of density = 50, 150, 250 veh/km.

**Figure 14 sensors-25-00302-f014:**
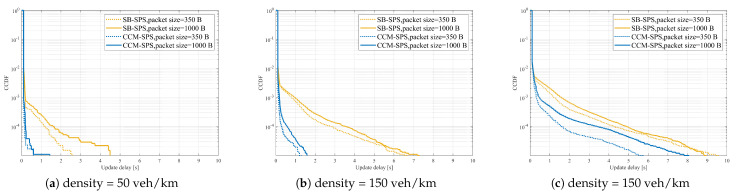
CCM-SPS’s communication performance evaluation on empirical CDF of update delay with different pack sizes in the case of density = 50, 150, 250 veh/km.

**Table 1 sensors-25-00302-t001:** Simulation parameters.

Parameter	Symbol	Value
Scenario		
Road layout	- -	Highway, 3 + 3 lanes
Density	- -	50, 150, 250 vehicles/km
Average speed	- -	70 km/h
STD of vehicle speed	- -	7 km/h
Target RCS	σ	10 dBsm
Power and propagation		
Channel model(interference)	- -	WINNER+, B1
Available channel bandwidth	$W_{ch}$	40 MHz
Transmitted power	$P_{T}$	23 dBm
Antenna gain	*G*	3 dBm
Noise figure	*F*	6 dB
Center frequency	$f_{c}$	5.9 GHz
Shadowing	- -	Variance 3 dB, decorr.dist. 25 m
Physical layer		
SCS	Δf	15 kHz
MCS	- -	5 (QPSK, $R_{c}=0.3$)
Sbuchannel size	- -	10 PRBs
Access layer		
Keep probability	$P_{keep}$	0.8
Initial reselection counter	RC	[5,15]
RSRP sensing threshold	- -	−126 dBm
Data traffic		
Packet generation interval	- -	100 ms
Packet size	- -	350, 1000 bytes

**Table 2 sensors-25-00302-t002:** Impact of parameters on occupied bandwidth.

Packet	SCS	MCS	$N_{\mathrm{PRB}}^{\max}$	$N_{\mathrm{sub}}$	$N_{\mathrm{PRB}}$	W [MHz]
350	15	5	216	4	40	7.2
1000	15	5	216	10	100	18

## Data Availability

Data are contained within the article.

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
