# Peer review of "Reinforcement Learning-Based Resource Allocation Scheme of NR-V2X Sidelink for Joint Communication and Sensing"

_sensors, 2025, doi:10.3390/s25020302_

Round 1

Reviewer 1 Report

Comments and Suggestions for Authors

The manuscript addresses a pertinent and contemporary topic, focusing on Joint Communication and Sensing (JCS) in NR-V2X systems utilizing reinforcement learning. While the technical content is robust, there are several areas where the paper could be strengthened to enhance its overall quality and impact:

  1. Motivation: The paper should provide a stronger rationale for the study by explicitly identifying the gaps in existing research. While the authors briefly mention the issue of consecutive collisions, more detailed discussion is needed to explain why this problem remains unresolved, particularly in high-density scenarios.
  2. Comparison with Existing Work: The authors should clearly articulate how their proposed method distinguishes itself from and outperforms existing solutions. Currently, the related work section largely cites previous studies without critically analyzing how this work addresses limitations or offers novel contributions.
  3. Coexistence with Other Technologies: The paper highlights the potential benefits of coexistence for improved spectral efficiency, a topic well-documented in the literature. The authors should explicitly differentiate their work from the following key studies and clarify their unique contributions:
    • Wild, T., Braun, V., & Viswanathan, H. (2021). Joint design of communication and sensing for beyond 5G and 6G systems. IEEE Access, 9, 30845-30857.
    • Masouros, C., Zhang, J.A., Liu, F., Zheng, L., Wymeersch, H., & Di Renzo, M. (2023). Guest editorial: Integrated sensing and communications for 6G. IEEE Wireless Communications, 30(1), 14-15.
    • Ibrahem, L.N., et al. (2023). Best relay selection strategy in cooperative spectrum sharing framework with mobile-based end user. Applied Sciences, 13(14), 8127.
    • Liu, M., Yang, M., Zhang, Z., et al. (2023). Sensing-Communication Coexistence vs. Integration. IEEE Transactions on Vehicular Technology, 72(6), 8158-8163.
    • Alhulayil, M., & López-Benítez, M. (2019). Static contention window method for improved LTE-LAA/Wi-Fi coexistence in unlicensed bands. In 2019 International Conference on Wireless Networks and Mobile Communications (WINCOM), 1-6.
  4. Interpretation of Results: The simulation results are presented effectively; however, the practical implications of these findings require more detailed discussion. Explicitly outline how the results translate into actionable insights or advancements for NR-V2X systems in real-world scenarios.
  5. Conclusion: While the conclusion provides a good summary of the study, it would benefit from a brief discussion of the study’s limitations and suggestions for future research directions. This addition would offer readers a more comprehensive understanding of the work's scope and potential for further development.

Author Response

Comments 1: Motivation: The paper should provide a stronger rationale for the study by explicitly identifying the gaps in existing research. While the authors briefly mention the issue of consecutive collisions, more detailed discussion is needed to explain why this problem remains unresolved, particularly in high-density scenarios.

Response 1: We sincerely appreciate your valuable feedback regarding the lack of motivation in our article. Recognizing the importance of this aspect, we have thoroughly revised the manuscript to include relevant description.

  • We have added the discussion on motivation in the introduction section Page 2, as follows :“SB-SPS scheme firstly senses the channel quality and select the available candidate resources and not selects a new resource for the next transmission until the retransmission counter (RC) return to zero, in order to avoid somehow packet collision during the sidelink communication access. There are also numerous studies concerning the modified SPS schemes to improve low latency communication [7][8]. But the existing researches have rarely studied the impacts of packet collision for sidelink sensing performance. According to the theoretical model of Cramér-Rao Lower Bound (CRLB), the resultant SINR of the echo due to collisions will significantly impact the sensing performance of JCS sidelink. Especially, in high-density scenarios, the probability of selecting the same resource for different vehicles quickly rises, resulting in deterioration of both communication and sensing performances. In fact, to obtain the knowledge of the dynamic echo channel state of JCS is still a big challenge and difficult to be resolved. So, it is rational for this paper to study the consecutive collision problem related to echo together with transmission signals of JCS sidelink.”

Comments 2: Comparison with Existing Work: The authors should clearly articulate how their proposed method distinguishes itself from and outperforms existing solutions. Currently, the related work section largely cites previous studies without critically analyzing how this work addresses limitations or offers novel contributions.

Response 2: Thank you for your insightful review and valuable feedback on our previous submission. We have revised the article to more clearly articulate the contributions of the cited works and have added further critical analyses on the value of CRLB theoretical foundation, limitation of traditional SB-SPS, and motivations from reinforce learning enhanced methods and collision detection methods with full-duplex.

  • In the third paragraph of the Related Work section, we added discussions:

         Deleted “Like other wireless access technologies in vehicular networks, NR-V2X also relies on                 OFDM at the physical (PHY) layer.”

         Add “In recent years, researches on the sensing capabilities of NR-V2X sidelink signals has been           increasingly explored. Studies on the theoretical foundation of CRLB for the sidelink sensing                 performance highlight the potential value for the resource allocation scheme of JCS sidelink                 system.”

         Add “The above existing research contributions can be referenced to make the theoretical                     performance analysis on sidelink JCS system in this paper.”

  • In the fourth paragraph of the Related Work section, we added discussions:

         Add “But in highly dynamic scenarios, there are uncertain complicated environment changes of             the network will intensify the packet collisions so that the above methods could not be adaptive           to solve the communication performance degradation.”

         Add “To overcome this problem, recent studies have increasingly focused on applying                           reinforcement learning techniques to optimize V2X resource allocation”

         Add “It is worthwhile to note from existing studies that the optimal RC parameter trained by RL             can intelligently reduce consecutive collision probabilities in the highly dynamic sidelink                       communication. Similarly, viewing that the RC optimization is crucial to sensing CRLB, this paper           will introduce RL approach to the RC optimization in the JCS sidelink resource allocation to                   improve sensing performance in the highly-dense scenario.”

  • In the fifth paragraph of the Related Work section, we added discussions: “Additionally, sidelink collision detection which indicates the dynamic change on channel states can be feasible through full-duplex technology.”

Comments 3: Coexistence with Other Technologies: The paper highlights the potential benefits of coexistence for improved spectral efficiency, a topic well-documented in the literature. The authors should explicitly differentiate their work from the following key studies and clarify their unique contributions:

Response 3: Thank you for your approval and suggestions, we fully acknowledge the need to more explicitly outline the benefits of coexistence and have cited the above suggested articles in [9]-[13]. Their unique contributions are clarified as below.

  • We have added a paragraph in the Related Work section to explain the advantages of coexistence:” Signal coexistence for achieving higher spectral efficiency has recently garnered significant attention, e.g. in [9]-[13]。Reference [9] proposes a system that combines collaborative communication technology with cognitive radio, which improves the overall performance of the system by cooperative spectrum sharing networks. Reference [10] explored the coexistence of LAA and WIFI over unlicensed bands through static contention window method. Their work provides practical solutions for achieving technological coexistence in existing network architectures. Reference [11] explores the advantages of integrating sensing and communication technologies (ISAC), including high spectral efficiency, low hardware costs, and improved system performance, as well as the potential for ISAC in future 6G applications. Reference [12] proposed a joint design framework for communication and sensing in small cellular networks, which optimizes the collaborative work of communication and sensing through waveform selection and resource allocation. Reference [13] addresses the practical challenges of residual hardware impairments (RHI) and imperfect successive interference cancellation, deriving the superior performance of the ISAC framework compared to the sensing-communication coexistence (SCC) framework. It demonstrates that the integration of sensing and communication is a significant trend for future development.”

Comments 4: Interpretation of Results: The simulation results are presented effectively; however, the practical implications of these findings require more detailed discussion. Explicitly outline how the results translate into actionable insights or advancements for NR-V2X systems in real-world scenarios.

Response 4: We appreciate your careful assessment of our manuscript. In order to strengthen the discussion of the NR-V2X system in real-life scenarios, we will implement the following modifications:

  • At the beginning of Section 6, we have added a discussion on the impact of performance metrics on practical applications. Specifically, “Among them, CRLB is a sensing performance metric that affects target detection accuracy, thereby influencing autonomous driving decision-making; PRR is a communication reliability metric that impacts communication quality; and UD is a latency metric that affects the real-time performance of intelligent transportation systems.”
  • At the end of Section 6.2, we have added a discussion on the practical significance of the experiments. Specifically, “The results demonstrate that the proposed CCM-SPS scheme outperforms the comparison methods in terms of CRLB and PRR across various density scenarios. There advantages have strong application potential in fields of high accuracy For example, in high-density environments, it maintains a CRLB accuracy of 0.1m and a reliable communication range of 50m, ensuring both accurate target sensing and reliable communication, which are critical for autonomous driving decision-making.”
  • At the end of Section 6.3, we add the expression: “Therefore, the CCM-SPS algorithm is capable to support dynamic data traffic service scenarios such as V2X network in intelligent transportation system."

Comments 5: Conclusion: While the conclusion provides a good summary of the study, it would benefit from a brief discussion of the study’s limitations and suggestions for future research directions. This addition would offer readers a more comprehensive understanding of the work's scope and potential for further development.

Response 5: Thank you very much for your thorough review and insightful comments, which have helped us improve the clarity and comprehensiveness of our manuscript. Your keen observation is much appreciated. We made the following revisions in the Conclusions section, further addressing the limitations of the study and outlining directions for future research.

  • Add “Besides V2X network studied in this paper, there is still room to explore CCM-SPS scheme to be used in various JCS applications, such as AIOT network. "

Reviewer 2 Report

Comments and Suggestions for Authors

This paper presents a reinforcement learning-based resource allocation scheme for NR-V2X sidelink in joint communication and sensing (JCS) systems. The proposed scheme aims to efficiently utilize signal echoes for both communication and object localization, supporting applications like Advanced Driver Assistance Systems (ADAS) and autonomous vehicles. The paper explores the application of NR-V2X sidelink in low-latency, high-reliability communication and optimizes resource allocation using reinforcement learning to enhance system performance. It is a topic of interest to the researchers in the related areas, but the paper needs some improvements at this stage. My detailed comments are as follows:

1. The paper presents a promising reinforcement learning-based resource allocation scheme for NR-V2X sidelink. However, the novelty compared to existing methods should be more clearly emphasized.

2. The description of the reinforcement learning approach is too brief. A more detailed explanation of the algorithm, training process, and simulation setup would strengthen the paper.

3. Some sections, especially the introduction, are dense and need clearer, more concise explanations. The flow between sections can also be improved for better readability.

4. The potential real-world impact is intriguing, but further discussion on scalability and integration with emerging technologies like edge computing would be valuable.

5. The related article explores the impact of inertia control on frequency stability in wind power systems and proposes methods for maintaining stability in extreme conditions. This insight can be applied to NR-V2X systems, where intelligent algorithms like reinforcement learning can optimize resource allocation, dynamically adjust communication and sensing resources, and ensure stability and low latency in varying network conditions, for example: https://doi.org/10.3390/su16124965.

6. The related article considers the uncertainty of electricity prices in its model, which is similar to the network load fluctuations and data traffic variations in the NR-V2X sidelink system. In a reinforcement learning-based resource allocation scheme, the system needs to handle these uncertainties, such as changes in network traffic and fluctuations in traffic density. By adopting similar methods and integrating reinforcement learning, resource allocation can be optimized in dynamic environments, ensuring the stability and efficiency of communication and sensing tasks, which can be found at: DOI: 10.1109/TCE.2024.3412803.

Author Response

Comments 1: The paper presents a promising reinforcement learning-based resource allocation scheme for NR-V2X sidelink. However, the novelty compared to existing methods should be more clearly emphasized.

Response 1: Your feedback has been invaluable in highlighting numerous areas of shortages within our origin manuscript. We have made the following modifications to address this issue.

  • In the beginning of fifth paragraph of the Related Work section, we revised a discussion. Modify “Further research incorporates methods such as reinforcement learning[30][31][32]” to “Recent studies have increasingly focused on applying reinforcement learning techniques to optimize V2X resource allocation[30][31][32]. Existing resource allocation methods often rely on static models, which are unable to adapt in real-time to the dynamic traffic environment. In contrast, reinforcement learning can autonomously adjust, significantly improving resource utilization and system performance.”

Comments 2: The description of the reinforcement learning approach is too brief. A more detailed explanation of the algorithm, training process, and simulation setup would strengthen the paper.

Response 2: Thank you for your insightful suggestions. We have supplemented the reinforcement learning at the end of Section 5.2.1 to illustrate the training  process.

  • This paper adopts an ε-greedy strategy to balance the exploitation-exploration process. During the training process, the initial ε set to 0.9 and gradually decreased to 0.1. The training lasted for 100 seconds, consisting of 1000 iterations. The initial learning rate was set to 0.01, and the discount factor was set to 0.9.

Comments 3: Some sections, especially the introduction, are dense and need clearer, more concise explanations. The flow between sections can also be improved for better readability.

Response 3: We modify the introduction section : clear expression, highlight focus, add context connection between sections, and make better organization.

Comments 4: The potential real-world impact is intriguing, but further discussion on scalability and integration with emerging technologies like edge computing would be valuable.

Response 4: Thank you for your insightful review and valuable feedback. We accept the suggestion and make the following modifications.

  • At the end of the Conclusions section, we add future prospect: “Additionally, the integration of edge computing with CCM-SPS scheme can further enhance the performance of the Sidelink JCS system to support rich and abroad JCS application tasks.”

Comments 5: The related article explores the impact of inertia control on frequency stability in wind power systems and proposes methods for maintaining stability in extreme conditions. This insight can be applied to NR-V2X systems, where intelligent algorithms like reinforcement learning can optimize resource allocation, dynamically adjust communication and sensing resources, and ensure stability and low latency in varying network conditions, for example: https://doi.org/10.3390/su16124965.

Response 5: We appreciate your valuable suggestions add the article in [10][11].

  • We have added a discussion on the above literature in the Introduction section. Add “Moreover, in response to the high dynamics of V2X traffic density and network load, inspired by references [10][11], reinforcement learning and other intelligent algorithms can be employed to optimize resource allocation, ensuring the stability and accuracy of communication and sensing tasks.”

Comments 6: The related article considers the uncertainty of electricity prices in its model, which is similar to the network load fluctuations and data traffic variations in the NR-V2X sidelink system. In a reinforcement learning-based resource allocation scheme, the system needs to handle these uncertainties, such as changes in network traffic and fluctuations in traffic density. By adopting similar methods and integrating reinforcement learning, resource allocation can be optimized in dynamic environments, ensuring the stability and efficiency of communication and sensing tasks, which can be found at: DOI: 10.1109/TCE.2024.3412803.

Response 6: We appreciate your valuable suggestions add the article in [10][11].

  • We have also added a discussion on the above literature in the fifth paragraph of the Related Word section. Add “Moreover, in response to the high dynamics of V2X traffic density and network load, inspired by references [10][11], reinforcement learning and other intelligent algorithms can be employed to optimize resource allocation, ensuring the stability and accuracy of communication and sensing tasks.”